## [Editor Report · Decision Letter 0]

15 May 2025

Dear Dr Yang,

Thank you for submitting your manuscript entitled "USP7-mediated deubiquitination and nuclear translocation of PFKM promotes brain tumor development by sensing fructose 2,6 bisphosphate" for consideration as a Research Article by PLOS Biology.

Your manuscript has now been evaluated by the PLOS Biology editorial staff, as well as by an academic editor with relevant expertise, and I am writing to let you know that we would like to send your submission out for external peer review.

Once your full submission is complete, your paper will undergo a series of checks in preparation for peer review. After your manuscript has passed the checks it will be sent out for review. To provide the metadata for your submission, please Login to Editorial Manager (https://www.editorialmanager.com/pbiology) within two working days, i.e. by May 17 2025 11:59PM.

Kind regards,

Taylor

Taylor Hart, PhD,

Associate Editor

PLOS Biology

thart@plos.org

---

## [Decision Letter · Decision Letter 1]

14 Jul 2025

Dear Dr Yang,

Thank you for your patience while your manuscript "USP7-mediated deubiquitination and nuclear translocation of PFKM promotes brain tumor development by sensing fructose 2,6 bisphosphate" was peer-reviewed at PLOS Biology. Your manuscript has been evaluated by the PLOS Biology editors, an Academic Editor with relevant expertise, and by several independent reviewers.

As you will see in the reviewer reports, which can be found at the end of this email, although the reviewers find the work potentially interesting, they have also raised a substantial number of important concerns. Based on their specific comments and following discussion with the Academic Editor, it is clear that a substantial amount of work would be required to meet the criteria for publication in PLOS Biology. However, given our and the reviewer interest in your study, we would be open to inviting a comprehensive revision of the study that thoroughly addresses all the reviewers' comments. Given the extent of revision that would be needed, we cannot make a decision about publication until we have seen the revised manuscript and your response to the reviewers' comments. Your revised manuscript would need to be seen by the reviewers again, but please note that we would not engage them unless their main concerns have been addressed.

As you will see, both reviewers find the conclusions interesting, however they also raise concerns about the strength of support provided for the mechanism proposed. These concerns relate to areas requiring further investigation and validation, the need for better links between the mouse and clinical data, and important information missing. The reviewers suggest experiments that should be performed to strengthen the mechanistic insights.

We appreciate that these requests represent a great deal of extra work, and we are willing to relax our standard revision time to allow you 6 months to revise your study. Please email us (plosbiology@plos.org) if you have any questions or concerns, or envision needing a (short) extension.

**IMPORTANT - SUBMITTING YOUR REVISION**

*Resubmission Checklist*

*Published Peer Review*

*PLOS Data Policy*

*Blot and Gel Data Policy*

Sincerely,

Taylor

Taylor Hart, PhD,

Associate Editor

PLOS Biology

thart@plos.org

REVIEWS:

Reviewer #1: Wu et al. report significant findings on the role of PFKM in glioblastoma under glucose-deficient conditions. They propose that USP7 deubiquitinates PFKM at lysine 615, enabling its translocation to the nucleus, where it interacts with c-MYC to upregulate CPT1B and activate fatty acid oxidation (FAO) for tumor cell survival. The study, supported by in vitro and in vivo experiments, highlights a correlation between nuclear PFKM levels and glioblastoma prognosis, suggesting that USP7 inhibitors may represent a potential therapeutic strategy. However, concerns about the experimental design, data interpretation, and presentation need to be addressed to strengthen the manuscript.

Major Concerns

1. The study claims that nuclear PFKM interacts with c-MYC to upregulate CPT1B expression, driving FAO. However, this interaction is not sufficiently supported. The manuscript should include evidence to support that the PFKM-c-MYC interaction indeed happens in the nucleus.

2. The study suggests that decreased F-2,6-BP levels under GD enhance the USP7-PFKM interaction, but the mechanism by which F-2,6-BP inhibits this interaction is unclear. Is F-2,6-BP directly binding to PFKM or USP7, or is it modulating their interaction indirectly (e.g., through conformational changes or other intermediates)

3. Functional relevance of FAO activation:

* Lines 197-205, the analysis is not objective.

* While the study demonstrates that nuclear PFKM upregulates CPT1B to activate FAO, it does not sufficiently consider the importance of other metabolic pathways for tumor cell survival under GD. These need to be experimentally tested or further discussed.

* The manuscript should include experiments inhibiting FAO (e.g., using etomoxir, a CPT1 inhibitor) to assess its necessity for tumor cell survival in PFKM-rescued cells under GD.

4. Clinical Correlation Data:

* The correlation between nuclear PFKM levels and GBM malignancy/prognosis is compelling, but the sample size (55 patients for survival analysis, 63 and 60 for low- and high-grade gliomas) is relatively small. Data from the TCGA or other public databases could be analyzed to provide additional support.

* The scoring system for nuclear PFKM staining (0-4.9 vs. 5-8) is arbitrary and lacks justification. How this threshold was determined needs more explanation.

5. The use of P22077 as a USP7 inhibitor shows promising results in reducing tumor growth and extending survival in mice. However, P22077's specificity for USP7 is not well-established, and off-target effects may contribute to the observed outcomes. More information and limitations about this inhibitor should be discussed.

Minor Comments

* There are some typographical errors in the manuscript.

* Lines 113-115, the authors need to better explain how they arrive at the mono-ubiquitination, but no other forms of PTMs were involved, in a more objective manner.

Reviewer #2 [Gabriel Leprivier]: S. Wu et al. investigated the molecular function of nuclear Phosphofructokinase, muscle type (PFKM) and its contribution to glioblastoma tumorigenesis. The authors found that nuclear translocation of PFKM is enhanced under glucose starvation to protect glioblastoma cells against this stress. Such a nuclear shuttling is regulated by mono-ubiquitination of PFKM which is controlled by the interaction of the USP7 deubiquitinase with PFKM, as the authors reported. Mechanistically, fructose-2,6-bisphosphate restricts the interaction between PFKM and USP7, and PFKM interacts with c-MYC to promote CPT1B transcription, which parallels an increase of fatty acid oxidation under glucose starvation. In vivo, the authors reported that a non-monoubiquitinated mutant of PFKM, which accumulates in the nucleus, promotes glioblastoma growth in an orthotopic model and that treatment with a USP7 inhibitor restricts glioblastoma tumor growth. Clinically, the authors showed that high levels of nuclear PFKM is a factor of poor prognosis in glioblastoma.

The study is of potentially interest for the readership of PLOS Biology, pending that a few major points get addressed:

- Figure 2: the authors should show how PFKM mono-ubiquitination prevents PKFM nuclear translocation. Among others, is mono-ubiquitination preventing NLS exposure?

- Figure 2 f-h: it would be important to show the levels of expression of PFKM WT and PFKM K615R in the glioblastoma tumors to compare their respective expression.

- Figure 3 e, f: these data are about the role of USP7 in glioblastoma tumorigenesis and may not be related to PFKM at all. So either the authors can show that the effect of the USP7 inhibitor relies on PFKM, otherwise the data in their current state should be removed from the manuscript in my opinion.

- Figure 5: the authors should check whether PFKM WT enhances CPT1B promoter activity (using a CPT1B promoter Luciferase reporter for instance). It would be also important to show whether PFKM K615R binds to the CPT1B promoter, especially under glucose starvation. And if it doesn't appear to be the case, it would be informative to investigate how PFKM K615R enhances c-MYC binding to the CPT1B promoter under glucose starvation

Minor points:

- Figure 1g, h: the authors should show representative images for the nuclear staining of PFKM, one for low staining, one for high staining (fig. 1g), and one for Grade II and one for Grade IV glioma (fig. 1h).

- Figure 5 a: the concentrations of Etomoxir and BPTES used is not mentioned anywhere. Also, where Etomoxir and BPTES were purchased is not mentioned.

- Few spelling mistakes need to be fixed: "-deificient" (p6, line 119), "Etomoxior" (fig 5a), "promotor" (fig 5h).

---

## [Decision Letter · Decision Letter 2]

4 Feb 2026

Dear Dr Yang,

Thank you for your patience while we considered your revised manuscript "USP7-mediated deubiquitination and nuclear translocation of PFKM promotes brain tumor development by sensing fructose 2,6 bisphosphate" for publication as a Research Article at PLOS Biology. This revised version of your manuscript has been evaluated by the PLOS Biology editors, the Academic Editor, and one of the original reviewers. Thanks also for your help in clarifying the change of authorship.

Based on the review and on our Academic Editor's assessment of your revision, we are likely to accept this manuscript for publication. Please also make sure to address the following data and other policy-related requests.

IMPORTANT: Please ensure that you address all of these editorial requirements, as they are required before we can formally accept your manuscript:

**Title:

We would like to modify your paper's title to provide more detail on the metabolic context and the identity of PFKM. As the study is clinically-relevant but primarily based on mouse experiments, we also would like to mention the main model species in the title. Is the following alternative version of the title acceptable to you?

“USP7 facilitates brain tumor survival upon glucose deprivation by regulating phosphofructokinase muscle-type nuclear translocation in mice"

**Financial disclosure statement:

-- Please add links to the funding agencies in the Financial Disclosure statement in the manuscript details.

**Ethics:

-- The Ethics statement needs to be a separate, independent (and the first) subheading in the Material & Methods section. It must include ethical approval information related both to the patient samples and the animal studies. It must include all relevant approval numbers, as well as the It must include the full name of the IACUC and ethics committee that reviewed and approved the experiments. Research involving human participants must have been conducted according to the principles expressed in the Declaration of Helsinki. https://journals.plos.org/plosbiology/s/ethical-publishing-practice

**Data:

Please supply the numerical values either in a supplementary excel file or as a permanent DOI’d deposition for the following figures panels:

1FJ

2EG

3DF

4D

5ABDEFGJ

S2D

S3BC

S4AB

S5ABDEF

Note: the numerical data provided should include all replicates AND the way in which the plotted mean and errors were derived (it should not present only the mean/average values).

-- Please cite the location of the data clearly in all relevant main and supplementary Figure legends, e.g. “The data underlying this Figure can be found in S1 Data” or “The data underlying this Figure can be found in https://doi.org/10.5281/zenodo.XXXXX”

-- Supplementary files (e.g., excel). Please ensure that all data files are uploaded as 'Supporting Information' and are invariably referred to (in the manuscript, figure legends, and the Description field when uploading your files) using the following format verbatim: S1 Data, S2 Data, etc. Multiple panels of a single or even several figures can be included as multiple sheets in one excel file that is saved using exactly the following convention: S1_Data.xlsx (using an underscore).

-- Please change the Data Availability Statement in the manuscript text to reflect the location of all data. Please note that making the data 'available upon reasonable request' does not meet our requirements.

**Supplement:

-- Please upload the Supplementary Figures as individual Supporting Information files. Please include all tables in the main manuscript document.

**Gels:

-- We require the original, uncropped and minimally adjusted images supporting all blot and gel results reported in the Figures:

1ABE

2ABCD

3ABC

4ABCEFGHI

5CHI

S1AC

S2ABCEFHIJK

S5C

-- We will require these files before a manuscript can be accepted so please prepare and upload them now. Please carefully read our guidelines for how to prepare and upload this data: https://journals.plos.org/plosbiology/s/figures#loc-blot-and-gel-reporting-requirements

**Species in abstract:

-- Please note that per journal policy, the model system/species studied should be clearly stated in the abstract of your manuscript.

We expect to receive your revised manuscript within two weeks.

*Published Peer Review History*

*Press*

Sincerely,

Taylor

Taylor Hart, PhD,

Associate Editor

thart@plos.org

PLOS Biology

REVIEW

Reviewer #2 [Gabriel Leprivier]: The authors convincingly addressed all my points.

---

## [Editor Report · Decision Letter 3]

24 Feb 2026

Dear Dr Yang,

Thank you for the submission of your revised Research Article "USP7 facilitates brain tumor survival upon glucose deprivation by regulating phosphofructokinase muscle-type nuclear translocation in mice" for publication in PLOS Biology. On behalf of my colleagues and the Academic Editor, Elena Rainero, I am pleased to say that we can in principle accept your manuscript for publication, provided you address any remaining formatting and reporting issues. These will be detailed in an email you should receive within 2-3 business days from our colleagues in the journal operations team; no action is required from you until then. Please note that we will not be able to formally accept your manuscript and schedule it for publication until you have completed any requested changes.

PRESS

Sincerely,

Taylor

Taylor Hart, PhD,

Associate Editor

PLOS Biology

thart@plos.org